# Proximate Analysis and Nutritional Evaluation of Twenty Canadian Lentils by Principal Component and Cluster Analyses

**DOI:** 10.3390/foods9020175

**Published:** 2020-02-11

**Authors:** D. Dan Ramdath, Zhan-Hui Lu, Padma L. Maharaj, Jordan Winberg, Yolanda Brummer, Aileen Hawke

**Affiliations:** Guelph Research and Development Center, Agriculture and Agri-Food Canada, Guelph, ON N1G 5C9, Canada; zhanhui.lu@canada.ca (Z.-H.L.); pmaharaj@mail.uoguelph.ca (P.L.M.); jwinberg@uoguelph.ca (J.W.); yolanda.brummer@canada.ca (Y.B.); aileen.hawke@canada.ca (A.H.)

**Keywords:** lentil, starch digestibility, hydrolysis curve, resistant starch, dietary fiber

## Abstract

Proximate composition and starch nutritional properties of twenty cooked lentils were assessed to identify unique varieties that could be used in value added foods. Significant variations exist among the lentil varieties (*p* < 0.05) with respect to their energy, fat, protein, carbohydrate, and dietary fiber content, and these are related to lentil type and seed size. Dazil and Greenstar were unique for their high resistant starch content (RS) and lower area under the starch hydrolysis curve (SHAUC) while Proclaim was opposite. SHAUC was positively correlated (*p* < 0.001) with rapidly digestible starch (RDS) content (*r* = 0.626) but negatively correlated with RS content (*r* = −0.635). Principal component analysis showed that the first three principal components accounted for 62.8% of the total variance and the contribution of SHAUC was 33.2%. These results confirm that in vitro SHAUC and a combination of RDS and RS may be predictive of the digestibility profile of cooked lentils.

## 1. Introduction

Pulses such as lentils (*Lens culinaris*) are nutrient-dense foods that are included in dietary recommendations and several clinical practice guidelines [1]. On average, dry lentil seeds (per 100 g) contain 65 g total carbohydrate, 20 g protein, 4 g fiber, and 0.6 g fat, being rich in several essential minerals [2]. Lentils are an excellent source of amino acids that can complement cereal proteins and are low in antinutritional factors [3]. Although Canada is the world’s leader in lentil production and export, this pulse has limited consumption and only recently started to gain popularity in the North American diet [4]. One of the reasons is the lack of properly developed food forms, which requires in-depth understanding of the chemical and nutritional properties of lentils [5].

Lentils contain a high amount of glycemic or “available” carbohydrates, which are classified into three main fractions of nutritional importance: rapidly digestible starch (RDS), slowly digestible starch (SDS), and resistant starch (RS) [6]. RDS is digested in the small intestine and leads to a rapid rise in blood glucose levels following ingestion. SDS, as an intermediate starch fraction between RDS and RS, is digested slowly throughout the entire small intestine to provide sustained and prolonged glucose release with a low initial glycemia. SDS is likely associated with positive health effects including glycemic control, reduced postprandial circulated free fatty acids, and reduced oxidative stress [7]. RS has the ability to modulate postprandial blood-glucose levels; it evades enzymatic hydrolysis in the small intestine, passing into the colon where it is fermented to produce butyrate-rich short chain fatty acids for energy [8]. In comparison with traditional fibers, RS provides better appearance, texture, and mouthfeel, thus offering better sensory properties and consumer acceptability of the final products [9]. Although starch fractions have been characterized for a few lentil varieties, it would be useful to have this information for a larger number of available varieties in order to better guide choices for optimal formulation of lentil based foods.

Foods with low GI typically contain high levels of SDS and RS and have been recommended for the treatment and prevention of diabetes, heart disease, and related conditions [10]; as such, lentil-based foods could make an important contribution. It is reasonable to assume that there will be variations in starch digestion profiles, and thus glycemic responses, among samples from different lentil types and varieties. This information can be obtained using in vitro digestion methods and provides an opportunity to assess the relationship between starch hydrolysis rate and predictive in vivo blood glucose responses using well-accepted in vitro protocols [11].

This study was undertaken to define the chemical composition, dietary fiber fractions and starch nutritional properties of current and emerging Canadian lentil varieties, and to identify unique lentil varieties and key factors affecting starch digestion in cooked lentils. Furthermore, dietary fiber fractions including galacto-oligosaccharides (GOS) were also investigated to advance the understanding of their influence on starch digestion. Finally, principal component analysis and cluster analysis were used to assess independent variables that contribute to lentil starch digestion.

## 2. Materials and Methods

### 2.1. Materials and Reagents

Twenty lentil varieties of red and green types and with varying seed sizes were obtained from the Saskatoon Crop Development Centre (CDC) in March 2014, and all the analyses were completed in 2015. The names and market classifications of each variety are listed in Table 1. Potato amylose (A0512), waxy corn starch (S9679), porcine pancreatin (P7545), pepsin (P7125) and invertase (I4504) were purchased from Sigma-Aldrich (St. Louis, MO, USA). Amyloglucosidase (EC 3.2.1.3, 3260 U/mL), total starch kits (K-TSTA), resistant starch kits (K-RSTAR), glucose oxidase-peroxidase kits (K-GLUC), and integrated total dietary fiber kits (K-INTDF) were purchased from Megazyme International Ireland Ltd. (Bray, Ireland). All other chemicals used were of analytical grade. Purified water was used and prepared using a NANOpure^®^ Infinity Ultrapure Water System (Barnstead, Dubuque, IA, USA).

### 2.2. Sample Preparation

Whole lentil seeds (150 g) were rinsed and cooked in a rice cooker (Model RC3406C Type 1, Black & Decker) with distilled water (1:3 seed:water ratio) as previously outlined [11]. The cooking time was in a range from 35 to 70 min depending on the variety. Tests for doneness began at 10 min after boiling. The time required to attain a soft product was determined based on a tactile evaluation method. Lentils were considered cooked when four out of five seeds had little to no resistance to squeezing between thumb and forefinger. Cooked lentils were cooled to room temperature, then freeze-dried at −40 °C (0.1 mBar; FreeZone 12, Labconco, Kansas, MO, USA) and subsequently ground (IKA M20 Universal mill; IKA^®^ Works, Inc., Wilmington, NC, USA) to pass through a 250 µm sieve. The ground samples were stored in sealed plastic bags at room temperature prior to analysis.

### 2.3. Chemical Analysis

Freshly cooked lentils were sent to a commercial service provider (University of Arkansas, Fayetteville, AR, USA) for proximate analysis, where standard AOAC (Association of Official Analytical Chemists) methods for total fat (AOAC 922.06), ash (AOAC 923.03), energy (bomb calorimetry) and protein (AOAC 992.15) were used. Total dietary fiber was analyzed in house (AOAC 991.43) using a commercial kit (K-TDFR-100A; Megazyme International Ireland Ltd. Bray, Ireland) with modifications to allow for the analysis of GOS. The filtrate for soluble dietary fiber (SDF) analysis was reduced in volume using a rotary evaporator (Büchi Rotavapor R-215) and diluted with deionized water in order to determine the content of nondigestible GOS by HPLC with an anion-exchange column and a pulsed amperometric detection unit (HPAEC-PAD) (Dionex, Sunnyvale, CA, USA) [12]. Raffinose, stachyose, and verbascose content were quantified by comparing with appropriate standards. Carbohydrates (CHO, %) were calculated as the difference (CHO% = [100 − moisture% – protein% – fat% – ash%]). Total starch content was determined in duplicate using AACC (American Association of Cereal Chemists) method 76-13.01 (AACC, 1999) with DMSO pretreatment as outlined by the kit manufacturer (Megazyme International Ireland Ltd., Bray, Ireland). Apparent amylose content was measured using an iodine colorimetric protocol [13].

### 2.4. In Vitro Starch Digestion

The method of Englyst et al. [14] with minor modifications was used to characterize the in vitro starch digestibility of cooked lentil samples. Porcine pancreatic α-amylase (0.45 g) was dispersed in water (4 mL), and centrifuged at 1500 × *g* for 12 min. The supernatant (2.7 mL) was transferred to a beaker, and amyloglucosidase (0.3 mL) and invertase (0.2 mL) were added to the solution. This enzyme solution was freshly prepared. For each digestion, 100 mg of cooked lentil sample (see Section 2.2 above) was added to 4 mL of 0.5 M sodium acetate buffer (pH 5.2) per test tube. The enzyme solution (1 mL) and 15 glass beads (4 mm diameter) were added and incubated in a shaking water bath (37 °C, 200 strokes/min). Aliquots (0.1 mL) were taken at 20 and 120 min and mixed with 1 mL of 80% ethanol, after which free glucose content was measured by the glucose oxidase-peroxidase reagent. Percentages of rapidly digestible starch (RDS, % digestible starch at 20 min), slowly digestible starch (SDS, % digestible starch at 120 min – % digestible starch at 20 min), and resistant starch (RS, 100% – % digestible starch at 120 min) were normalized to the total starch content. For comparison, direct quantification of RS and nonresistant starch (nonRS) was performed in triplicate using a Megazyme RS assay kit (K-RSTAR, Megazyme International Ireland Ltd., Bray, Ireland). Briefly, 100 mg of cooked lentil sample was incubated with pancreatic α-amylase (10 mg/mL) containing amyloglucosidase (3 U/mL) in 4 mL of sodium acetate butter (1.2 M, pH 3.8) at 37 °C with continuous shaking (200 strokes/min) for exactly 16 h. After incubation, 4 mL of ethanol (99%) was added to inactivate the enzyme and the sample was centrifuged at 1500 × *g* for 10 min. Glucose content of the supernatant was measured by a glucose oxidase-peroxidase assay kit (Megazyme International Ireland Ltd., Bray, Ireland). NonRS was starch digested within 16 h and RS was starch not hydrolyzed even after 16 h. To differentiate between the RS obtained from Englyst et al.’s method and Megazyme kit, the term “RS-direct” was used for the RS obtained by the direct assay kit.

### 2.5. Estimated Glycemic Index (eGI)

The digestion kinetics of the lentil samples were tracked by a nonlinear model established by Goni et al. [15]. The first-order equation is $C=C_{\infty }\left(1-e^{-kt}\right)$. The area under the in vitro starch hydrolysis curve (SHAUC, mg min mL^−1^) was calculated from the integrals of equation $\mathrm{SHAUC}=C_{\infty }\left(t_{f}-t_{0}\right)-\frac{C_{\infty }}{k}\left[1-e^{-k\left(t_{f}-t_{0}\right)}\right]$ from 0 (*t*_0_) to 120 min (*t_f_*), where *C* is the percentage of starch hydrolyzed at time *t* (min), *C*_∞_ is the equilibrium percentage of starch hydrolyzed after 120 min, and *k* is the kinetic constant. The hydrolysis index (HI) was calculated as the ratio of the area under the hydrolysis curve (0–120 min) of lentil sample and the area of standard material from white bread. The expected glycemic index (*e*GI) was calculated using equation (*e*GI = 8.198 + 0.862 × HI) proposed by Granfeldt et al. [16].

### 2.6. Statistical Analysis

Samples were tested at least in triplicate. Statistical analyses (*t*-test and one-way analysis of variance (ANOVA)) were performed using SAS (Version 9.3 for Windows, SAS Institute Inc., Cary, NC, USA). When appropriate, the difference amongst means was determined using Tukey’s *post hoc* test, except that Dunnett’s method was used for multiple comparison of kinetic parameters and *e*GI of cooked lentils to those of white bread control. Pearson correlation coefficients and principal component analysis (PCA) were performed on centered and standardized data to elucidate the relationships among variables of the chemical and nutritional properties of samples. Cluster analysis (CA) was conducted using furthest neighbor linkage method and with Euclidean distance being the measure of similarity. Dendrogram and *K*-means plot were drawn to visualize the clusters of lentil samples. Both PCA and CA were conducted by OriginLab 9 (OriginLab Corporation, Northampton, MA, USA). Statistical significance was set at the 5% level of probability.

## 3. Results and Discussion

### 3.1. Proximate Analysis of Cooked Lentils

Results of proximate analyses of the twenty freeze-dried and ground cooked lentils are shown in Table 1. The range of values (100 g dry weight) for the lentil varieties were: 1843 kJ (Improve) to 1885 kJ (Roxy) for energy; 2.70% (Impower) to 3.74% (Greenstar) for ash; 0.50% (Viceroy, Rosie) to 1.49% (Roxy) for fat; 26.5% (Proclaim) to 31.6% (Kermit) for protein; and 64.2% (Kermit) to 69.1% (Proclaim) for carbohydrate (Table 1). These values are in a similar range to the previous studies (different lentil varieties) [17,18] and varied among the lentil types, seed size, and varieties. The energy values were obtained by bomb calorimetry, and hence the lower energy content of fiber and resistant starch were not taken into account.

On average, red lentils had higher fat content (*p <* 0.01) and lower carbohydrate and total starch content (*p <* 0.05) than green-type lentils, regardless of seed size. At the same seed size, fat content in red type (1.2%) was significantly higher than that of green type (0.9%) (*p <* 0.05). Within red-type lentils, the extra-small seeds showed higher ash and protein content (*p <* 0.001) but lower fat (*p <* 0.05) and lower carbohydrate (*p <* 0.01) content than the small seed. This observation was also true for green-type lentils, i.e., small seeds also had higher protein content and lower carbohydrate content (*p <* 0.05) (Table 1). The variations of proximate compositions are likely to result in different nutritional properties of cooked lentils.

### 3.2. Dietary Fiber Composition of Cooked Lentils

Dietary fiber composition results for the cooked lentils are presented in Table 2, with the following range of values: 14.1% (Imigreen) to 16.5% (Proclaim) for insoluble dietary fiber (IDF); 1.6% (Impress) to 2.6% (Impala) for SDF; 16.0% (Imigreen) to 18.6% (Proclaim) for total dietary fiber (TDF); 0.19% (Greenland, Dazil) to 0.50% (Impala) for raffinose; 1.91% (Improve) to 3.35% (Scarlet) for stachyose; and 0.47% (Asterix) to 1.96% (Impower) for verbascose. All DF ranged from 19.8% (Imigreen) to 22.6% (Proclaim). These ranges are similar to those obtained for different lentil varieties in other studies [17,19,20], with acceptable variations from lentil type, variety, growing year, location and sample preparation methods. In contrast to the proximate analysis results, smaller variations were seen in dietary fiber compositions among cooked lentil samples. On average, red-type lentils had higher dietary fiber fractions (IDF, SDF, TDF, stachyose, and all DF) but lower verbascose than that of green type, regardless of seed size (*p <* 0.05). At the same seed size, verbascose in red type (1.1%) was significantly higher than that of green type (0.9%) (*p <* 0.05). Within red-type lentils, extra-small seeds showed higher SDF and raffinose content but lower verbascose content than small seeds (*p <* 0.01). Within green-type lentils, small and medium size seeds had significantly lower verbascose content than large size seed (*p <* 0.05). Interestingly, verbascose was higher in larger seed-size lentils regardless of lentil type (Table 2).

### 3.3. Total Starch, Apparent Amylose Content and In Vitro Starch Digestibility Determined by Indirect and Direct Methods

Total starch content, apparent amylose content, and starch nutritional fractions determined by the method of Englyst et al. [14] (indirect method) and Megazyme RS kit (direct method) are shown in Table 3. The total starch ranged from 42.2% (Kermit) to 45.7% (Impress), which is in accordance with the typically reported starch content of lentil flours [18,21]. A larger variation was seen with apparent amylose content, which ranged from 7.9% (Scarlet) to 12.4% (Improve). Lu et al. [5] previously reported 19–26% of apparent amylose in eight raw Canadian lentil flours. It is likely that cooking resulted in amylose complexing with other lentil constituents (e.g., lipids and phenolic compounds) and/or, less likely, that amylose may have leached out into cooking water, thereby accounting for the lower amylose content. The reduction in amylose content after cooking was also observed in cooked rice [22].

Using the indirect method [14], RDS, SDS, and RS content of the cooked lentils ranged from 73.5% (Redberry) to 85.5% (Impower), 1.9% (Dazil) to 14.4% (Redberry), and 8.0% (Proclaim) to 16.7% (Dazil), respectively. The starch fractions measured by Megazyme RS assay kit were in the ranges of 88.1% (Kermit) to 97.5% (Imvincible) for nonRS and 2.5% (Imvincible) to 11.9% (Kermit) for RS-direct, respectively. The results of RS amount determined by the method of Englyst et al. [14] are in accordance with one previous study [5], but higher than values reported by Costa et al. [17] and lower than those by others [15,23,24,25]. Moreover, the RS values measured by the direct method were generally lower than those measured by the indirect method, especially for Proclaim, Imvincible, and Greenstar (Table 3). These inconsistencies have been attributed to factors such as sample preparation, pretreatment, and incubation conditions (stirring or shaking, pH, temperature, time, and enzyme mixtures) inherent to different methods [26,27,28] and to the nature of plant materials. Consistent RS values were reported for bean flakes, corn flakes, and canned beans by either the Englyst method or Megazyme kit, but large variations were observed on native potato starch and native and retrograded amylomaize starch [28].

Within a specific method, e.g., the method of Englyst et al. [14], variations in starch digestibility may be attributed to individual starch characteristics of different lentil varieties and interactions (e.g., matrix effect, complexing) between starch and other chemical compositions in cooked lentils, such as protein [29], lipids [30], and phenolic compounds [31,32]. Besides starch–phenoliccompound complexing, which increases SDS and RS amount, flavonols in lentils were also reported to be responsible for the inhibitory activities against α-glucosidase [32]. Therefore, its role on various starch fractions of these twenty cooked lentils should not be ignored.

As shown in Table 3, the average RS content was ~11% for either red-type lentils or green-type lentils, with no significant difference. However, Lu et al. [5] reported that, on average, cooked red lentil flour had significantly higher RS content (11.0%) than flour from green lentils (6.8%) (*p <* 0.05) and attributed the difference to higher total flavonol index of red lentils (452.5 μg g^−1^ db) than that of green lentils (449.9 μg g^−1^ db). Another study showed that there were exceptions with two green lentils that contained high levels of phenolic compounds [32]. The inconsistency could be attributed to the extremely low RS content (8%) of Proclaim (red type), and high RS content (13.1%) of Greenstar (green type) in this set of lentil samples. These two samples appear to be outliers among these twenty lentils (Table 3). Cooking methods may also account for differences, as lentil seeds were cooked whole in this study, not heated in a test tube as a powder [5].

RS content obtained by Megazyme kit (RS-direct) supports the previous observations [5] and also showed that, on average, smaller seed-size lentils had higher RS content, regardless of lentil type (*p <* 0.05, Table 3). The above results suggest that starch digestibility of cooled lentils was more dependent on variety than type.

### 3.4. Kinetic Study of Starch Digestion and Estimated Glycemic Index (eGI)

Starch hydrolysis curves were plotted as time vs. percent digestible starch (derived from glucose measurements) taken at different sampling times (0, 5, 10, 20, 60, and 120 min) during in vitro digestion (Appendix A). The average percentage of starch hydrolyzed to glucose after 20 min of digestion was 82.3%. This is similar to the finding reported by Wong et al. [33]. Afterwards, an average increase of only 6.7% was observed from 20 to 60 min, indicating that the starch hydrolysis was nearing completion. Similarly, Zhou et al. [34] found that the plateau for in vitro hydrolysis of lentil starch occurred after about 85% of the starch had been digested. Additionally, there was very little difference in the amount of glucose released between 60 and 120 min of starch hydrolysis, and none of the lentils were fully hydrolyzed. The latter was also observed with gelatinized red lentil starch, in which 90.4% was fully hydrolyzed following digestion [35].

The kinetic parameters (*C*_∞_, *k*), SHAUC, HI and *e*GI for each lentil sample and for the reference food (white bread) are presented in Table 4. The range of values observed were: 81.6% (Dazil) to 90.3% (Proclaim) for *C*_∞_; 0.14 min^−1^ (Redberry) to 0.33 min^−1^ (Imvincible, Dazil) for *k*; 9527 (Redberry) to 10,516 (Proclaim) for SHAUC (mg min mL^−1^); 84% (Redberry) to 92.7% (Proclaim) for HI; and 81 (Dazil, Imvincible, Redberry) to 88 (Proclaim) for *e*GI. The *e*GI of cooked lentils obtained in this study is slightly higher than the GI of 74 for canned lentil listed in The International Table of Glycemic Index and Glycemic Load Values [36]. The difference could be attributed to the extensive sample preparation processes [37] including cooking, freeze-drying, and mechanical grinding using a cyclone mill in this study, which would expose starch to digestive enzymes because of much reduced particle size [38], increased available surface area for enzymatic action [39], and potential depolymerization of starch due to milling.

No significant differences were identified in the starch hydrolysis variables among lentil types or seed size; however, these were all significantly different from white bread control (Dunnett *post hoc*; *p <* 0.05). Because of collinearity among *e*GI, HI, and SHAUC, and the fact that SHAUC is a primary outcome of starch digestibility, SHAUC, instead of *e*GI, was used in multivariate analysis that explored key factors affecting lentil starch digestibility, and in identifying unique lentil varieties.

### 3.5. Pearson Correlation Between Chemical Components and Nutritional Properties

Pearson correlation coefficients between chemical components and nutritional properties of the cooked lentils are presented in Table 5. RDS was negatively correlated with energy (*p* = 0.023) and protein (*p* = 0.007) and was positively correlated with carbohydrate (*p* = 0.013) and stachyose (*p* = 0.025). SDS was negatively correlated with stachyose (*p* = 0.031). As expected, derived RS significantly correlated with these nutrients, but in an opposite direction to RDS. Of the results obtained using the Megazyme kit, only carbohydrate showed a significant (*p <* 0.05) correlation with nonRS and RS-direct. In contrast, SHAUC was significantly correlated with more measures of chemical composition and with most dietary fiber components, except SDF and verbascose. SHAUC was not correlated with carbohydrate content or total starch. A recent study with lentils also found that none of the individual starch fractions was significantly associated with total starch and total dietary fiber content [11]. As such, it appears that SHAUC is largely determined by apparent amylose and dietary fiber fractions.

Results from the Englyst method [14] showed good correlation with in vivo glycemic response using six meals, including lentil soup [40], and additional human studies have identified correlations between glycemic index (GI) and rapidly digestible starch of commonly consumed starchy foods [41]. However, a recent study showed that the low GI value of cooked lentils is not reliably predicted by the individual RDS, SDS, or RS values, but instead by SHAUC [11]. It is therefore suggested that SHAUC, rather that its derivative, *e*GI, be the primary index used to assess the overall digestion profile of cooked lentils.

### 3.6. Principal Component Analysis (PCA)

PCA provides an overview of the similarities and differences between the measured properties and was used in this study to visualize the variations in chemical components and nutritional properties of the cooked lentils. The results showed that the proportions of eigenvalues of the correlation matrix for the first three principal components were 28.5%, 22.0%, and 12.3%. Two components accounted for 50.5% of the total variance and three components explained 62.8%, indicating that three components could provide a good summary of the data. A biplot in which the points represent lentil varieties and the vectors represent chemical components and nutritional parameters of cooked lentils is presented in Figure 1.

The directions of the vectors indicate that PC1 explains starch digestion fractions determined by the Englyst method, dietary fibers, and SHAUC; PC2 distinguishes chemical compositions (e.g., carbohydrate, total starch, and protein, excluding dietary fibers) (Figure 1A); and PC3 appears to represent nonRS and RS-direct contents measured by Megazyme kit, showing almost no relation to SHAUC (Figure 1B). The pair of variables describing SHAUC, RDS and *C*_∞_, were highly correlated and also highly negatively correlated to RS, suggesting that RDS and RS determined by the indirect method [14] are good indicators of SHAUC and are derived within a similar timeframe (i.e., 120 min of digestion), rather than by 16 h of hydrolysis by using a Megazyme kit. Correspondingly, RS was grouped with energy, protein, and amylose, showing close correlations among them and a negative correlation with SHAUC. As expected, all dietary fiber parameters were highly correlated with each other and weakly correlated with RDS and RS (Figure 1), which is consistent with a previous study [11].

Lentil varieties that are clustered together in the biplot are those that have similar profiles of chemical and nutritional properties. Green lentils are mostly located in the middle to left panel of PC1, unique in lower dietary fiber content, while red lentils are located in the middle to upper panel of PC2 (except Proclaim), showing lower carbohydrate and total starch (Figure 1A). Within red-type lentils, Roxy, Dazil, and Redberry were closely grouped for their high amount of RS (Figure 1A, Table 3). Cherie and Imvincible are relatively far apart from them, being not only high in RS but also rich in SDS. Rosie, Scarlet, and Impulse were close together and characterized by high dietary fiber content. Proclaim is unique for its extremely high starch digestibility. Within the green-type lentils, Greenstar and Kermit showed similar properties to Dazil and Roxy. Greenland and Asterix are located near the center of biplot, which is indicative of intermediate characteristics. Impress, Viceroy, Improve, Imigreen, and Impower are far apart from other green lentil samples, owing to their specific profiles of total starch and carbohydrate.

In other studies, Asterix has been reported to be of low GI [11]. Asterix and Greenland have been reported unique for their high SDS content [5], owing to their high phenolic content and α-glucosidase inhibitory activity [32]. Although Asterix and Greenland did not stand out from the current sample set of 20 cooked lentils in PCA biplot (Figure 1), their RS content measured by Megazyme kit (RS-direct) was the second highest among the nine green lentils and was higher than 6 out of 11 red lentils (Table 3).

Pearson correlation and PCA were informative and showed that RDS, RS, and SHAUC are reliable indicators to differentiate the chemical compositions and nutritional properties of these twenty lentils. However, the biplot did not show clear separations of lentil varieties corresponding to starch digestibility and/or SHAUC. Therefore, cluster analysis was conducted to group these twenty lentils, specifically focusing on RS and SHAUC.

### 3.7. Cluster Analysis (CA)

A dendrogram from hierarchical cluster analysis and *K*-means plot corresponding to the RS content and SHAUC values of these twenty cooked lentils are shown in Figure 2A,B. The results suggest that twenty lentils could be divided into four groups based on the visual segmentation in Figure 2B, mainly by SHAUC:

Group 1, low SHAUC (<9825 mg min mL^−1^), high RS (10–17%); includes Dazil, Redberry, Imvincible, Greenstar, Cherie, Roxy, and Viceroy;

Group 2, intermediate SHAUC (9825–10,200 mg min mL^−1^), intermediate RS (9–13%); includes Greenland, Kermit, Impala, Impress, Imax, Improve, Imigreen, and Impower;

Group 3, high SHAUC (10,200–10,400 mg min mL^−1^), intermediate RS (9–13%); includes Rosie, Asterix, Scarlet, and Impulse; unique in overall higher dietary fiber content;

Group 4, the highest SHAUC (>10,400 mg min mL^−1^) and low RS (<9%); the variety is Proclaim.

Comparing the variety linkage in Figure 2A with variety grouping in Figure 2B, it was found that the variety linkage and grouping were clearly based more on SHAUC values than RS content. The reasons could be: (1) that there was a larger magnitude of SHAUC values than that of RS; and (2) that a combination of RDS and RS was more predictive than any individual starch digestible fraction [11]. Nonetheless, SHAUC was highly correlated to most chemical compositions (Table 5), successfully clustered the samples (Figure 2), and thus appears to be a primary indicator of nutritional properties of the cooked lentils.

## 4. Conclusions

The proximate composition, dietary fiber fractions, and starch nutritional properties were significantly different among the twenty cooked lentil varieties. The differences were more variety- than type-dependent. Generally, cooked red lentils tend to have lower GI values than green lentils. However, Proclaim was unique for its lowest RS while having the highest *e*GI. In contrast, Greenstar, which is a green-type lentil, had relatively higher RS, but a lower *e*GI, than most of the other lentils (*p <* 0.05). Regardless of lentil type, smaller seed size seems to be associated with higher RS content in cooked lentils. Dietary fiber fractions appeared to have little correlation with starch digestible fractions but were highly correlated with SHAUC. Based on SHAUC and RS content, twenty lentil varieties could be classified into four groups by PCA and CA. The consistency between PCA and CA indicated that multivariate analyses were feasible in the study of lentil characteristics and useful for selecting the appropriate variety for final use suitability.

## Figures and Tables

**Figure 1 foods-09-00175-f001:**
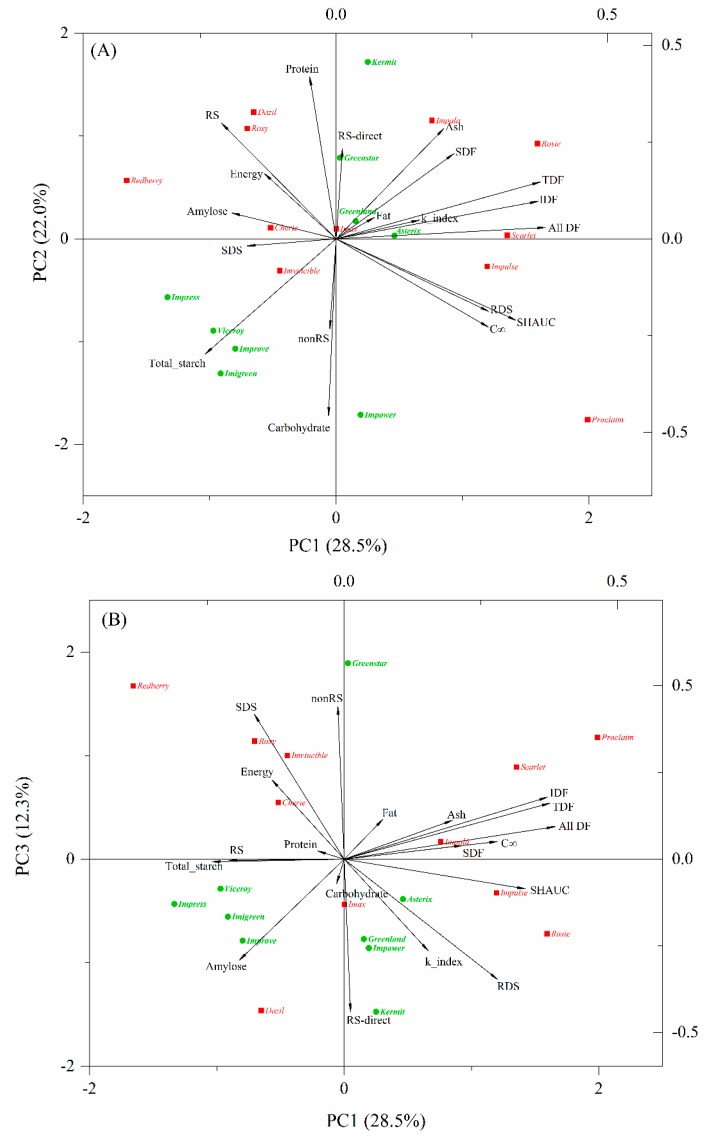
Biplot of the principal component analysis PC1 vs. PC2 (**A**) and PC1 vs. PC3 (**B**), describing the scores and variation in chemical and nutritional properties of cooked lentils. Abbreviations: SDF, IDF, TDF, and All_DF, soluble, insoluble, total and all dietary fibers, respectively; RDS, SDS and RS, rapidly digestible, slowly digestible and resistant starch, respectively, measured as outlined by Englyst et al.; nonRS and RS_kit, nonresistant and resistant starch, respectively, measured by Megazyme resistant starch kit; *C*_∞_, maximum hydrolysis extent; *k*_index, kinetic constant; SHAUC, the area under the in vitro starch hydrolysis curve.

**Figure 2 foods-09-00175-f002:**
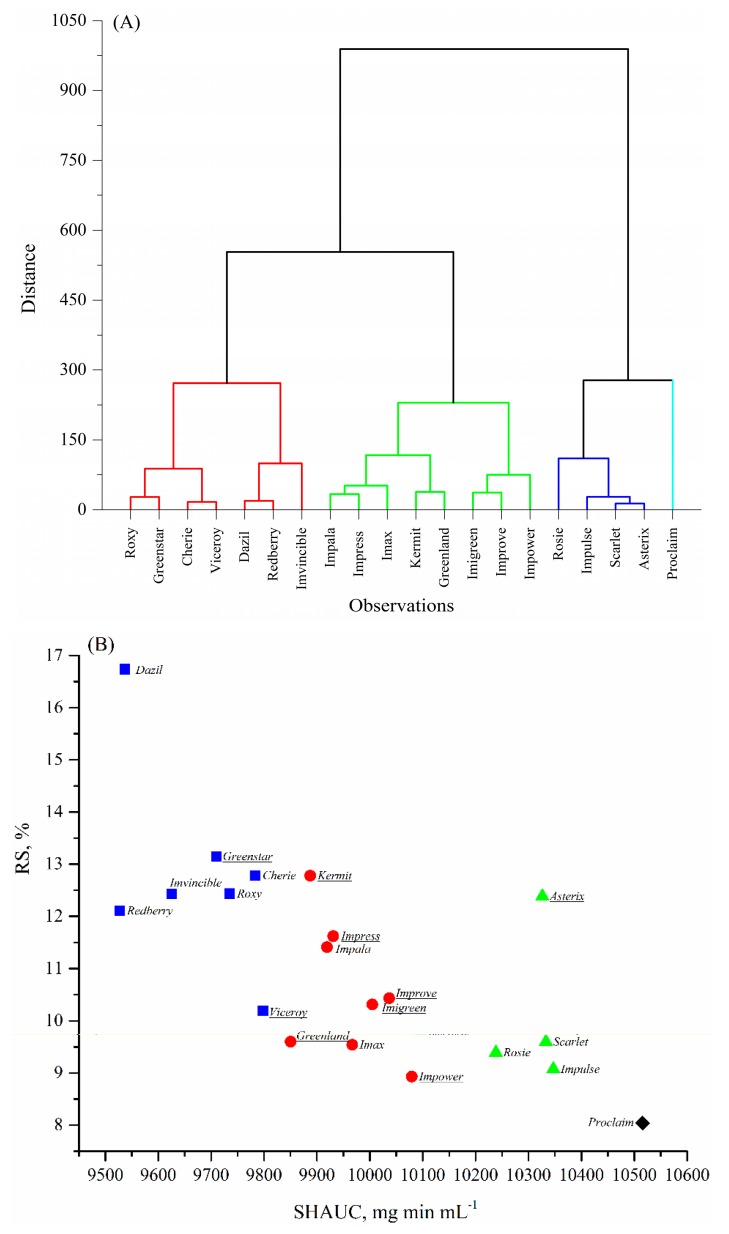
Cluster analysis of the 20 lentil varieties and four plotted groups corresponding to the area under the in vitro starch hydrolysis curve (SHAUC) and RS value. (**A**) Dendrogram by Hierarchical cluster analysis; (**B**) *K*-means plot corresponding to the SHAUC and RS values. Symbols in Figure 2B, ■, cluster 1; ●, cluster 2; ▲, cluster 3; and ♦, cluster 4. Green lentil names were underlined.

**Table 1 foods-09-00175-t001:** Proximate analysis of freeze-dried cooked lentil powders (dry weight basis).

**Color/Class/Variety**	**Energy, kJ**	**Ash, %**	**Fat, %**	**Protein, %**	**Carbohydrate, %**
Red lentils					
*Extra small*					
Roxy	1885 ± 6 a	3.13 ± 0.01 f	1.49 ± 0.04 a	30.4 ± 0.2 bcd	65.0 ± 0.2 ijk
Impala	1846 ± 1 i	3.43 ± 0.01 d	0.85 ± 0.06 h	31.2 ± 0.1 ab	64.5 ± 0.1 jk
Rosie	1872 ± 5 abcdef	3.58 ± 0.01 c	0.50 ± 0.03 j	30.5 ± 0.6 abc	65.4 ± 0.6 hij
*Small*					
Cherie	1875 ± 3 abcde	3.12 ± 0.02 f	1.25 ± 0.05 bc	29.4 ± 0.5 defgh	66.2 ± 0.5 fgh
Dazil	1877 ± 3 abcd	3.27 ± 0.01 e	1.25 ± 0.05 bc	29.3 ± 0.4 defgh	66.2 ± 0.4 fgh
Impulse	1854 ± 6 hi	3.03 ± 0.04 g	1.33 ± 0.00 b	28.9 ± 0.3 efgh	66.7 ± 0.4 defg
Proclaim	1862 ± 1 efgh	3.17 ± 0.01 f	1.25 ± 0.04 bc	26.5 ± 0.3 i	69.1 ± 0.3 a
Imax	1865 ± 3 defgh	3.26 ± 0.02 e	1.16 ± 0.03 cd	29.3 ± 0.4 defgh	66.3 ± 0.4 fgh
Imvincible	1881 ± 2 ab	3.01 ± 0.02 g	0.93 ± 0.03 fgh	29.6 ± 0.4 cdefgh	66.4 ± 0.4 efgh
Redberry	1879 ± 3 abc	2.89 ± 0.01 h	0.93 ± 0.03 fgh	30.5 ± 0.2 abc	65.7 ± 0.2 ghi
Scarlet	1877 ± 2 abcd	3.40 ± 0.02 d	1.13 ± 0.02 de	30.0 ± 0.5 cde	65.5 ± 0.5 hij
Green lentils					
*Small*					
Kermit	1869 ± 4 bcdef	3.16 ± 0.03 f	1.03 ± 0.05 ef	31.6 ± 0.2 a	64.2 ± 0.2 k
Asterix	1867 ± 5 cdefg	3.02 ± 0.01 g	1.30 ± 0.05 b	29.9 ± 0.3 cdef	65.8 ± 0.4 ghi
Viceroy	1880 ± 6 abc	3.00 ± 0.01 g	0.50 ± 0.02 j	28.9 ± 0.3 fgh	67.6 ± 0.3 cd
*Medium*					
Imigreen	1861 ± 2 fgh	2.74 ± 0.03 i	0.62 ± 0.05 i	28.8 ± 0.2 gh	67.8 ± 0.2 bc
Impress	1877 ± 6 abcd	2.97 ± 0.03 g	0.88 ± 0.02 gh	29.1 ± 0.4 efgh	67.1 ± 0.4 cdef
*Large*					
Greenland	1877 ± 7 abcd	3.64 ± 0.03 b	0.91 ± 0.02 gh	28.5 ± 0.5 h	66.9 ± 0.5 cdef
Greenstar	1878 ± 4 abcd	3.74 ± 0.01 a	0.97 ± 0.02 fg	29.7 ± 0.5 cdefg	65.6 ± 0.5 ghij
Impower	1856 ± 4 ghi	2.70 ± 0.01 i	1.10 ± 0.03 de	27.4 ± 0.3 i	68.8 ± 0.4 ab
Improve	1843 ± 5 i	2.84 ± 0.03 h	0.98 ± 0.01 fg	28.6 ± 0.3 gh	67.6 ± 0.3 cde

Values followed by a different letter in the same column are significantly different (*p* < 0.05).

**Table 2 foods-09-00175-t002:** Dietary fiber and galacto-oligosaccharides content of freeze-dried cooked lentil powders (%, dry weight basis).

**Color/Class/Variety**	**IDF**	**SDF**	**TDF**	**Raffinose**	**Stachyose**	**Verbascose**	**All DF**
Red lentils							
*Extra small*							
Roxy	15.1 ± 0.5 abcd	2.2 ± 0.2 abc	17.3 ± 0.3 abcd	0.31 ± 0.02 fgh	2.28 ± 0.00 ghi	0.71 ± 0.08 j	20.7 ± 0.2 bcd
Impala	15.3 ± 0.9 abcd	2.6 ± 0.1 a	17.9 ± 0.8 abc	0.50 ± 0.05 a	2.68 ± 0.10 bcd	0.72 ± 0.02 j	21.8 ± 0.9 abc
Rosie	16.1 ± 0.5 ab	2.4 ± 0.3 ab	18.5 ± 0.3 ab	0.42 ± 0.01 bcd	2.81 ± 0.02 b	0.69 ± 0.01 j	22.4 ± 0.3 a
*Small*							
Cherie	15.0 ± 0.4 bcd	2.0 ± 0.0 abc	17.0 ± 0.4 cd	0.32 ± 0.01 fgh	2.77 ± 0.04 bc	1.13 ± 0.02 fghi	21.2 ± 0.4 abcd
DazilDazil	14.9 ± 0.2 bcd	2.1 ± 0.3 abc	17.0 ± 0.4 cd	0.19 ± 0.00 i	2.60 ± 0.07 cde	1.16 ± 0.06 efgh	20.9 ± 0.6 abcd
Impulse	15.6 ± 0.3 abc	2.2 ± 0.1 abc	17.7 ± 0.3 abc	0.42 ± 0.01 bc	2.30 ± 0.05 gh	1.27 ± 0.02 cde	21.7 ± 0.3 abc
Proclaim	16.5 ± 0.1 a	2.2 ± 0.2 abc	18.6 ± 0.2 a	0.36 ± 0.02 cdef	2.48 ± 0.02 ef	1.22 ± 0.09 def	22.6 ± 0.2 a
Imax	15.1 ± 0.2 abcd	2.0 ± 0.2 abc	17.2 ± 0.3 abcd	0.33 ± 0.01 efgh	2.58 ± 0.08 def	1.30 ± 0.03 bcd	21.4 ± 0.4 abcd
Imvincible	15.0 ± 0.4 bcd	1.9 ± 0.2 abc	17.0 ± 0.5 cd	0.29 ± 0.01 gh	2.61 ± 0.08 cde	1.19 ± 0.04 defg	21.1 ± 0.5 abcd
Redberry	14.7 ± 0.2 cd	2.0 ± 0.3 abc	16.7 ± 0.5 cd	0.27 ± 0.04 h	2.14 ± 0.07 hij	1.01 ± 0.04 i	20.1 ± 0.6 cd
Scarlet	15.8 ± 0.2 abc	2.0 ± 0.1 abc	17.7 ± 0.2 abc	0.34 ± 0.03 efgh	3.35 ± 0.03 a	0.69 ± 0.00 j	22.1 ± 0.3 ab
Green lentils							
*Small*							
Kermit	15.5 ± 0.5 abcd	2.0 ± 0.2 abc	17.5 ± 0.7 abc	0.29 ± 0.00 gh	2.17 ± 0.04 hij	1.04 ± 0.06 hi	21.0 ± 0.8 abcd
Asterix	15.5 ± 0.3 abc	1.8 ± 0.3 bc	17.3 ± 0.3 abcd	0.32 ± 0.02 fgh	3.26 ± 0.01 a	0.47 ± 0.01 k	21.3 ± 0.3 abcd
Viceroy	14.5 ± 0.2 cd	1.9 ± 0.2 abc	16.4 ± 0.4 cd	0.35 ± 0.01 defg	2.40 ± 0.08 fg	1.08 ± 0.02 ghi	20.3 ± 0.4 cd
*Medium*							
Imigreen	14.1 ± 0.2 d	1.9 ± 0.3 abc	16.0 ± 0.1 d	0.37 ± 0.03 cdef	2.11 ± 0.01 ij	1.21 ± 0.06 defg	19.8 ± 0.2 d
Impress	14.8 ± 0.6 bcd	1.6 ± 0.1 c	16.4 ± 0.6 cd	0.30 ± 0.02 gh	2.20 ± 0.06 hij	1.01 ± 0.03 i	19.9 ± 0.6 d
*Large*							
Greenland	15.1 ± 0.9 bcd	2.0 ± 0.2 abc	17.0 ± 0.8 bcd	0.19 ± 0.01 i	2.60 ± 0.01 cde	1.37 ± 0.06 bc	21.1 ± 1.0 abcd
Greenstar	15.9 ± 0.5 abc	1.9 ± 0.3 abc	17.8 ± 0.6 abc	0.39 ± 0.01 bcde	2.08 ± 0.01 jk	1.40 ± 0.01 b	21.7 ± 0.6 abc
Impower	15.1 ± 0.7 abcd	1.8 ± 0.0 bc	16.9 ± 0.7 cd	0.43 ± 0.01 ab	2.85 ± 0.04 b	1.96 ± 0.03 a	22.2 ± 0.7 ab
Improve	14.5 ± 0.1 cd	2.0 ± 0.3 abc	16.5 ± 0.2 cd	0.37 ± 0.01 cdef	1.91 ± 0.04 k	1.21 ± 0.02 defg	20.0 ± 0.2 d

Values are presented as the mean ± SD, *n* = 3. Different letters following values within the same column indicate a statistically significant difference (*p* < 0.05). Abbreviations: IDF, insoluble dietary fiber; SDF, soluble dietary fiber; TDF, total dietary fiber (TDF = IDF + SDF); All DF = TDF + galacto-oligosaccharides (raffinose + stachyose + verbascose).

**Table 3 foods-09-00175-t003:** Lentil total starch, apparent amylose content, and starch fractions (%, dry weight basis).

**Color/Class/Variety**	**Total Starch**	**Apparent Amylose ***	**Indirect Assay (Protocol by Englyst et al.)**			**Direct Assay**	
			**RDS**	**SDS**	**RS**	**nonRS**	**RS-Direct**
Red lentils							
*Extra small*							
Roxy	44.7 ± 0.2 bcde	8.7 ± 0.0 fg	77.6 ± 0.3 f	10.0 ± 0.5 b	12.4 ± 0.2 bc	90.6 ± 1.0 fghi	9.4 ± 1.0 cdef
Impala	43.1 ± 0.3 ij	10.6 ± 0.9 bcde	81.9 ± 0.4 abcde	6.7 ± 1.0 bcd	11.4 ± 0.5 bcdef	92.2 ± 0.2 efg	7.8 ± 0.2 efg
Rosie	43.7 ± 0.5 fghi	11.0 ± 0.4 abcd	85.3 ± 0.2 a	5.3 ± 0.5 cde	9.4 ± 0.3 efg	88.4 ± 0.7 jk	11.6 ± 0.7 ab
*Small*							
Cherie	44.1 ± 0.2 efgh	12.3 ± 0.0 ab	80.3 ± 1.1 cdef	6.9 ± 0.9 bcd	12.8 ± 0.2 bc	94.4 ± 0.3 cd	5.6 ± 0.3 hi
Dazil	44.6 ± 0.5 cdef	11.4 ± 0.5 abc	81.4 ± 0.0 bcde	1.9 ± 0.9 e	16.7 ± 0.9 a	89.1 ± 0.3 ijk	10.9 ± 0.3 abc
Impulse	43.1 ± 0.4 ij	9.3 ± 0.5 efg	84.0 ± 0.8 abc	6.9 ± 0.8 bcd	9.1 ± 0.0 efg	90.2 ± 1.0 ghijk	9.8 ± 1.0 abcde
Proclaim	44.4 ± 0.2 defg	8.7 ± 0.3 fg	85.3 ± 0.0 a	6.6 ± 0.8 bcd	8.0 ± 0.8 g	96.1 ± 1.0 abc	3.9 ± 1.0 ijk
Imax	44.4 ± 0.4 defgh	12.0 ± 0.0 ab	82.0 ± 0.3 abcde	8.4 ± 0.2 bc	9.5 ± 0.2 defg	90.1 ± 0.6 ghijk	9.9 ± 0.6 abcde
Imvincible	44.9 ± 0.3 abcde	9.7 ± 0.4 cdef	81.3 ± 0.7 bcdef	6.3 ± 0.1 bcd	12.4 ± 0.5 bc	97.5 ± 0.3 a	2.5 ± 0.3 k
Redberry	44.5 ± 0.1 cdef	10.7 ± 0.7 bcde	73.5 ± 0.6 g	14.4 ± 0.6 a	12.1 ± 0.0 bcd	92.1 ± 1.0 efgh	7.9 ± 1.0 defg
Scarlet	43.1 ± 0.1 ij	7.9 ± 0.3 g	84.3 ± 0.8 ab	6.1 ± 1.8 bcde	9.6 ± 1.0 defg	93.5 ± 0.3 de	6.5 ± 0.3 gh
Green lentils							
*Small*							
Kermit	42.2 ± 0.4 j	12.0 ± 0.4 ab	83.3 ± 0.2 abcd	3.9 ± 0.1 de	12.8 ± 0.2 bc	88.1 ± 0.6 k	11.9 ± 0.6 a
Asterix	44.3 ± 0.2 defgh	9.0 ± 0.1 fg	82.4 ± 0.1 abcde	5.3 ± 0.9 cde	12.4 ± 1.0 bc	90.8 ± 0.9 fghi	9.2 ± 0.9 cdef
Viceroy	45.5 ± 0.8 abc	11.4 ± 0.1 abc	84.4 ± 0.4 ab	5.4 ± 1.2 cde	10.2 ± 0.8 cdefg	94.5 ± 0.0 bcd	5.5 ± 0.0 hij
*Medium*							
Imigreen	44.6 ± 0.5 cdef	9.5 ± 0.2 def	83.3 ± 1.4 abcd	6.3 ± 0.9 bcd	10.3 ± 0.5 cdefg	93.4 ± 0.3 de	6.6 ± 0.3 gh
Impress	45.7 ± 0.1 a	12.1 ± 0.4 ab	80.0 ± 0.7 def	8.4 ± 1.7 bc	11.6 ± 1.0 bcde	89.9 ± 0.2 hijk	10.1 ± 0.2 abcd
*Large*							
Greenland	43.6 ± 0.4 ghi	11.1 ± 0.5 abcd	84.7 ± 2.8 ab	5.7 ± 2.8 bcde	9.6 ± 0.0 defg	90.5 ± 0.3 fghij	9.5 ± 0.3 bcdef
Greenstar	43.4 ± 0.2 hi	10.9 ± 0.4 abcde	79.2 ± 1.5 ef	7.7 ± 0.1 bcd	13.1 ± 1.4 b	96.5 ± 0.2 ab	3.5 ± 0.2 jk
Impower	45.6 ± 0.2 ab	10.9 ± 0.0 abcde	85.5 ± 0.7 a	5.6 ± 0.0 cde	8.9 ± 0.6 fg	92.5 ± 0.3 def	7.5 ± 0.3 fgh
Improve	45.2 ± 0.2 abcd	12.4 ± 0.4 a	81.8 ± 0.9 abcde	7.8 ± 0.9 bcd	10.4 ± 0.0 cdefg	92.6 ± 0.5 def	± 0.5 fgh

***** Apparent amylose content is expressed on a dry weight basis as a percentage of total starch. Values are presented as the mean ± SD, *n* = 4. Values succeeded by a different letter in the same column are significantly different (*p* < 0.05). Abbreviations: RDS, rapidly digestible starch; SDS, slowly digestible starch; RS, resistant starch; nonRS and RS-direct, nonresistant starch and resistant starch obtained by the direct Megazyme RS assay kit (K-RSTAR).

**Table 4 foods-09-00175-t004:** Exponential model parameters (*C*_∞_, *k*), area under the curve (SHAUC), hydrolysis index (HI), and estimated glycemic index (*e*GI) of Canadian lentil varieties.

**Color/Class/Variety**	***C*_∞_, %**	***K*, min^−1^**	**SHAUC, mg min mL^−1^**	**HI, %**	***e*GI**
Control (white bread)	94.6 ± 0.8	2478.33 ± 43.41	11343 ± 94	100.0 ± 0.8	95 ± 1
Red lentils					
*Extra small*					
Roxy	82.5 ± 3.6 cd	0.19 ± 0.02 gh	9735 ± 145 bcde	85.8 ± 1.3 bcde	82 ± 1 bcde
Impala	86.3 ± 0.5 abcd	0.21 ± 0.01 defg	9919 ± 16 abcde	87.4 ± 0.1 abcde	84 ± 0 abcde
Rosie	88.8 ± 0.5 ab	0.24 ± 0.01 cde	10238 ± 51 abcd	90.3 ± 0.5 abcd	86 ± 0 abcd
*Small*					
Cherie	86.1 ± 1.8 abcd	0.21 ± 0.03 defg	9783 ± 56 bcde	86.2 ± 0.5 bcde	83 ± 0 bcde
Dazil	81.6 ± 0.4 d	0.33 ± 0.02 a	9537 ± 37 e	84.1 ± 0.3 e	81 ± 0 e
Impulse	89.0 ± 1.2 ab	0.23 ± 0.01 cdef	10347 ± 31 ab	91.2 ± 0.3 ab	87 ± 0 ab
Proclaim	90.3 ± 0.7 a	0.25 ± 0.01 bcd	10516 ± 10 a	92.7 ± 0.1 a	88 ± 0 a
Imax	86.2 ± 1.5 abcd	0.23 ± 0.01 cdef	9967 ± 151 abcde	87.9 ± 1.3 abcde	84 ± 1 abcde
Imvincible	85.7 ± 2.9 abcd	0.33 ± 0.00 a	9626 ± 104 de	84.9 ± 0.9 de	81 ± 1 de
Redberry	85.0 ± 2.4 abcd	0.14 ± 0.02 i	9527 ± 199 e	84.0 ± 1.8 e	81 ± 2 e
Scarlet	90.2 ± 1.5 a	0.24 ± 0.01 cdef	10333 ± 89 ab	91.1 ± 0.8 ab	87 ± 1 ab
Green lentils					
*Small*					
Kermit	84.5 ± 0.8 bcd	0.29 ± 0.01 ab	9887 ± 89 bcde	87.2 ± 0.8 bcde	83 ± 1 bcde
Asterix	87.7 ± 1.8 abc	0.29 ± 0.01 ab	10326 ± 183 abc	91.0 ± 1.6 abc	87 ± 1 abc
Viceroy	84.0 ± 3.6 bcd	0.21 ± 0.03 defg	9798 ± 395 bcde	86.4 ± 3.5 bcde	83 ± 3 bcde
*Medium*					
Imigreen	86.9 ± 1.1 abcd	0.21 ± 0.01 efg	10005 ± 15 abcde	88.2 ± 0.1 abcde	84 ± 0 abcde
Impress	87.2 ± 1.3 abc	0.15 ± 0.02 hi	9931 ± 133 abcde	87.6 ± 1.2 abcde	84 ± 1 abcde
*Large*					
Greenland	84.6 ± 2.4 bcd	0.27 ± 0.02 bc	9850 ± 260 bcde	86.8 ± 2.3 bcde	83 ± 2 bcde
Greenstar	83.7 ± 2.3 bcd	0.20 ± 0.01 fg	9710 ± 216 cde	85.6 ± 1.9 cde	82 ± 2 cde
Impower	86.5 ± 2.6 abcd	0.26 ± 0.01 bc	10079 ± 174 abcde	88.9 ± 1.5 abcde	85 ± 1 abcde
Improve	86.3 ± 1.4 abcd	0.21 ± 0.01 efg	10037 ± 28 abcde	88.5 ± 0.2 abcde	84 ± 0 abcde

Data are expressed as the mean ± SD, *n* = 4. Values followed by a different letter in the same column are significantly different (ANOVA followed by Tukey’s test (*p* < 0.05)); all lentil varieties showed significant difference to control (white bread) by (ANOVA followed by Dunnett test (*p* < 0.05)). Abbreviations: *C*_∞_, equilibrium percentage of starch hydrolyzed after 120 min; *k*, kinetic constant; SHAUC, area under the curve; HI, hydrolysis index; *e*GI, estimated glycemic index.

**Table 5 foods-09-00175-t005:** Pearson’s correlation coefficients (*r*) between chemical compositions and starch nutritional parameters of cooked lentils.

**Variables**	**Englyst et al.’s Method**			**Megazyme Kit**		**SHAUC**
	**RDS**	**SDS**	**RS**	**nonRS**	**RS-Kit**	
Energy	−0.377 *	0.148	0.398 *	0.145	−0.145	−0.381 *
Ash	0.167	−0.22	0.033	−0.029	0.029	−0.015
Fat	−0.15	0.088	0.12	−0.178	0.178	0.102
Protein	−0.441 **	0.104	0.560 ***	−0.328	0.328	−0.315 *
Carbohydrate	0.408 *	−0.068	−0.556 ***	0.335 *	−0.335 *	0.27
Total starch	−0.124	0.198	−0.068	0.239	−0.239	−0.188
Apparent amylose	−0.14	0.012	0.206	−0.049	0.049	−0.434 *
IDF	0.176	−0.112	−0.124	0.006	−0.006	0.495 ***
SDF	0.054	−0.009	−0.074	−0.145	0.145	0.268
TDF	0.177	−0.103	−0.139	−0.041	0.041	0.534 ***
Raffinose	0.208	−0.015	−0.311	0.226	−0.226	0.438 **
Stachyose	0.396 *	−0.381 *	−0.083	0.021	−0.021	0.370 *
Verbascose	0.114	−0.002	−0.18	0.241	−0.241	−0.113
All DF	0.3	−0.212	−0.186	0.052	−0.052	0.562 ***

*, ** and ***, correlations are significant at *p* < 0.05, 0.01 and 0.001, respectively. IDF, insoluble dietary fiber; SDF, soluble dietary fiber; TDF, total dietary fiber; All DF, TDF plus raffinose, stachyose and verbascose; SHAUC, area under the curve; *e*GI (estimated glycemic index) and HI (hydrolysis index) share exactly same coefficients with SHAUC because of collinearity.

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
