# Peer review of "Proximate Analysis and Nutritional Evaluation of Twenty Canadian Lentils by Principal Component and Cluster Analyses"

_foods, 2020, doi:10.3390/foods9020175_

Round 1

Reviewer 1 Report

The manuscript is well written and easy to read. Thank you! The introduction properly outlines the analyzed problem. The methodology is correct and up to date. The discussion of the data is correct and adequate. The conclusions drawn are beyond doubt. The literature is vast and comprehensive, however more than half was published over that 10 years ago. Despite the high scientific value of the manuscript, I would like to suggest several minor changes:

Line 16 please add word “content” to clarify

Line 25 please delete “about”, the statement about composition is already about average content

Line 27-28 please rephrase, this sentence is only properly understood by reading the next one

Line 34 please provide information regarding digestion of SDS, similarly as it was done in case of RDS and RS

Line 38 citation strongly recommended for above statement

Line 97 unnecessary repetition of each

Line 138 The variation among energy values is statistically significant, however the differences between extreme cases are below 3%.  I wouldn’t call it considerable for such energy dense food product, especially from consumer perspective.

Line 149 -150 Duplication of this information in not necessary.

Line 245-246 The depolymerisation of starch due to milling could also slightly emphasize this effect.

Line 260-278 The correlation coefficients values are quite low, please reconsider placing the supp table in text or discuss it more thoroughly.

Line 288-289 The statement regarding PC1 is doubtful.

Figure 1. There is little to no discussion regarding PC1 vs. PC3 plot, please consider placing a 3D projection of PC1 vs. PC2 vs. PC3 instead of current figure.

Line 331 “these” is not required

Sidenote: please consider K-Amyl kit for analysis of amylose content in further research. This method seems to be more accurate in most cases.

Reviewer 2 Report

The article “Proximate Analysis and Nutritional Evaluation of Twenty Canadian Lentils by Principal Component  and Cluster Analyses” by: D. Dan Ramdath , Zhan-Hui Lu, Padma L. Maharaj, Jordan Winberg, Yolanda Brummer and Aileen Hawke.

The most important notes:

Tables 1 to 4 are presented and described correctly and in a comprehensible manner. In this part of the work, I am asking for minor corrections:

1.) verse 74 - no description of the device used for freeze-drying

2.) verse 11 - please indicate the time intervals for sampling

3.) verse 105 - please provide the characteristics of the method used, the sample preparation, the composition of the reagent used. This information can be relevant when discussing differences in RS results obtained using different research methods.

In the further part of the work (verse 260 and following), the authors present statistical analysis of the results. The authors publish a very large number of calculations and mathematical analyzes of the results obtained. The description of this part is insufficient for the reader to follow the authors' thinking. The drawings shown are not accurately described. No explanation of all abbreviations used. Also, the presented conclusions do not justify the need to use such a number of calculations and statistical analyzes.  I recommend limiting the number of statistical analyzes presented and expanding their description. Please, make the presented statistical method treated as a tool to obtain specific conclusions from the research results.

Reviewer 3 Report

The manuscript focuses on the nutritional properties of different varieties of lentils originating in Canada. The work is interesting because of the number of compared varieties, thanks to which it is possible to estimate the average nutritional value of seeds of this legume species.

Page 2, line 70: The description of cooking conditions should be supplemented by the range of time used to cook different varieties, for example “from 5 to 15 min depending on the variety”.

Page 3, line 100: Amylase in vitro digestion was carried out at 37 °C. Digestion time was not specified, this should be completed. The same part of description specifies a speed of 200 strokes/min. Why was this rotation selected and thus wat was simulated?

Page 3, line 134: The energy value of lentils was about 1850 kJ. The value was determined on the basis of combustion in a calorimetric bomb. In this way, the lower energy content of fiber and resistant starch was not taken into account. The energy value should be calculated taking into account fiber or at least a comment to the presented values should be given.

Section 3.1. has no discussion, only results, similar in point section 3.2.

Table 2. Are the galacto-oligodaccharides a part of  soluble fiber?

Page 5: Did all varieties have a husk and whether possible lack of husk could have an effect on the leaching of ingredients, including starch.

Page 8, line 263-264. The authors write about the occurrence of positive or negative correlation, while the occurrence of a significant correlation can be stated in the case of an absolute correlation coefficient value p > 0.4.

Reviewer 4 Report

The manuscript aims to evaluate the proximate analyses and the nutritional quality of 20 different cooked lentils. Most of the manuscript is focused on the dietary fiber and starch analysis, the last of which analyzed with two methods.
The used methods are correct and data seem reliable. Honestly, data in table are quite difficult to be understood from the reader, as for the huge amount of numbers and for the several letters of the statistical analysis, however food science data have this dark side. If there is a better method for the explanation, this would be appreciated.
Main comments:
- On the whole, it can be assumed that there are no substantial differences among the variety of lentils. Despite the two PCs account for 50% of the variance (and this is not so high, in fact there is a third PC), each one of the PC has a similar load, and it can be noticed that they cannot be highlighted close groups of lentils (above all in PC1/PC2 plot). However, in the PCA plots it is not
taken into consideration the size of the different lentils, but just the color. It should be changed just the symbol within the red or green lentils, so that the size can be shown. The same for the analysis in Figure 3.
Furthermore, it can be of help to split the load and score plot into two different plots for a better visualization.
- Table 1: it can be of help to furnish as supplementary tables 1 and 2 but with the statistical analysis performed within each “green” or “red lentils”, in order to know which one is “the best” for its characteristics.
- A comment for authors: can authors hypothesize with a small paragraph in the discussion whether the soaking of such lentils overnight (which is a practice made in different countries) might affect the results, so mainly the effect on starch?
- Paragraph 2.1: do authors have information about the growing conditions of the lentils? Agronomic conditions?
- Reference 1 should refer to guidelines, and reference 2 to a database. Please check.

Round 2

Reviewer 2 Report

Thank you for taking my comments into account and improving the text.

Reviewer 3 Report

The manuscript may be considered in the current version.

Reviewer 4 Report

Authors replied to this reviewer's queries. I have no further comments.